# Acute ampakines increase voiding function and coordination in a rat model of SCI

Sabhya Rana[1,2,3†], Firoj Alom[4,5†], Robert C Martinez[1,2,3], David D Fuller[1,2,3], Aaron D Mickle[2,4,5,6*]

[1]Department of Physical Therapy, University of Florida, Gainesville, United States; [2]McKnight Brain Institute, University of Florida, Gainesville, United States; [3]Breathing Research and Therapeutics Center, Gainesville, United States; [4]Department of Physiological Sciences, College of Veterinary Medicine, University of Florida, Gainesville, United States; [5]Department of Veterinary and Animal Sciences, University of Rajshahi, Rajshahi, Bangladesh; [6]J. Crayton Pruitt Family Department of Biomedical Engineering, College of Engineering, University of Florida, Gainesville, United States

*For correspondence: amickle@ufl.edu

†These authors contributed equally to this work

Competing interest: The authors declare that no competing interests exist.

**Abstract** Neurogenic bladder dysfunction causes urological complications and reduces the quality of life in persons with spinal cord injury (SCI). Glutamatergic signaling via AMPA receptors is fundamentally important to the neural circuits controlling bladder voiding. Ampakines are positive allosteric modulators of AMPA receptors that can enhance the function of glutamatergic neural circuits after SCI. We hypothesized that ampakines can acutely stimulate bladder voiding that has been impaired due to thoracic contusion SCI. Adult female Sprague–Dawley rats received a unilateral contusion of the T9 spinal cord (n = 10). Bladder function (cystometry) and coordination with the external urethral sphincter (EUS) were assessed 5 d post-SCI under urethane anesthesia. Data were compared to responses in spinal-intact rats (n = 8). The 'low-impact' ampakine CX1739 (5, 10, or 15 mg/kg) or vehicle (2-hydroxypropyl-beta-cyclodextrin [HPCD]) was administered intravenously. The HPCD vehicle had no discernible impact on voiding. In contrast, following CX1739, the pressure threshold for inducing bladder contraction, voided volume, and the interval between bladder contractions were significantly reduced. These responses occurred in a dose-dependent manner. We conclude that modulating AMPA receptor function using ampakines can rapidly improve bladder-voiding capability at subacute time points following contusion SCI. These results may provide a new and translatable method for therapeutic targeting of bladder dysfunction acutely after SCI.

## eLife assessment

Bladder dysfunction following spinal cord injury (SCI) represents a severe and disabling complication without effective therapies. Following evidence that AMPA receptors play a key role in bladder function, the authors show **convincingly** that AMPA allosteric activators can ameliorate many of the subacute defects in bladder and sphincter function following SCI, including prolonged voiding intervals and high bladder pressure thresholds for voiding. These **valuable** results in rodents may help in the development of these agents as therapeutics for humans with SCI-induced bladder dysfunction.

## Introduction

Traumatic spinal cord injury (SCI) often results in bladder dysfunction, which can lead to bladder infection and reduced quality of life (*Hamid et al., 2018*; *Piatt et al., 2016*). Indeed, restoration of bladder

function is ranked as one of the highest priorities by individuals with SCI (*Bourbeau et al., 2020*; *French et al., 2010*). The fundamental problem is uncoordinated voiding due to damage to sensory and motor circuits that control bladder coordination. The symptoms can include urinary retention (hypoactive) or incontinence (hyperactive) depending on the injury and time after recovery, leading to urodynamic complications such as urinary infection, kidney damage, bladder cancer, urethral strictures, and bladder stones (*Taweel and Seyam, 2015*).

The synaptic pathways that serve to coordinate the activation of bladder detrusor and external urethral sphincter (EUS) muscles rely on glutamatergic neurotransmission, and in SCI, these circuits can be disrupted (*de Groat et al., 2015*; *Fowler et al., 2008*; *Yoshiyama, 2009*; *Yoshiyama et al., 1999*). The α-amino-3-hydroxy-5-methyl-4-isoxazolepropionic acid (AMPA) glutamate receptor plays a significant role in mediating drive in the intact micturition reflex (*Yoshiyama and de Groat, 2005*; *Yoshiyama et al., 1997*). After SCI, the bladder is initially in an areflexive state, and as spinal circuits are reorganized, the bladder transitions to neurogenic detrusor overactivity (*de Groat et al., 2015*; *de Groat and Yoshimura, 2006*). Glutamatergic signaling and AMPA receptors are altered during these states (*Grossman et al., 2001*; *Grossman et al., 1999*; *Mitsui et al., 2011*; *Pikov and Wrathall, 2001*; *Pikov and Wrathall, 2002*). In general, there are decreases in AMPA receptor subunits initially after an injury during the hypoactive bladder state (*Grossman et al., 1999*).

The AMPA receptor can be allosterically modulated using pharmacological compounds called ampakines (*Arai and Kessler, 2007*), and recent work suggests that these compounds have therapeutic value after SCI (*Rana et al., 2021*; *Wollman et al., 2020*). For example, when given after acute or chronic SCI in rats, ampakines have a powerful, beneficial impact on the glutamatergic synaptic pathways, which control breathing (*Rana et al., 2021*; *Wollman et al., 2020*). After high-cervical SCI, diaphragm output is diminished, but systemically providing a low dose of ampakine can cause a rapid and sustained increase in diaphragm electromyogram output (*Rana et al., 2021*). Since depolarization of the phrenic motoneurons that innervate the diaphragm critically depends on AMPA receptor activation (*Chitravanshi and Sapru, 1996*; *Fuller et al., 2022*; *Rana et al., 2020*), ampakines are likely acting at least in part on spinal circuits to stimulate diaphragm output (*Thakre et al., 2022*).

Building on the foundation of prior work in ampakine treatment for respiratory insufficiency, we studied the impact of ampakines on voiding reflexes after SCI. An adult rat model of subacute (5 d) thoracic contusion SCI was used to test the hypothesis that intravenous delivery of ampakine CX1739 would have a dose-dependent ability to restore bladder and EUS functions and their coordinated voiding. Bladder function was evaluated by continuous flow cystometry (*Andersson et al., 2011*). The results demonstrate a remarkable ability of ampakine treatment to restore voiding reflexes following SCI. Ampakines' rapid and powerful impact on bladder function provides a foundation for a new and translatable method for therapeutic targeting of bladder dysfunction after SCI.

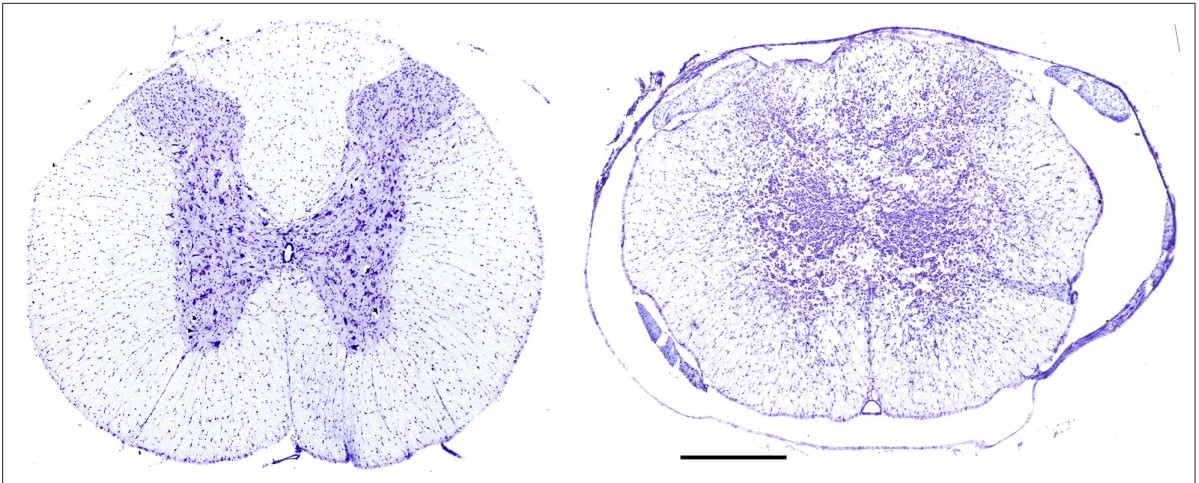

**Figure 1.** Histological assessment of T9 contusion (T9Ct) injury epicenter. Serial, transverse sections (20 µm thickness) spanning ~1 cm around the lesion epicenter (T9–T10) were stained with cresyl violet in order to identify the extent of injury. A representative image from a spinal intact (left) and T9 contused (right) spinal cord. Scale bar, 300 µm.

# Results

## T9 contusion injury

A T9 midline contusion injury was effectively delivered in 10 rats, with a distinct hematoma visually evident under the surgical microscope. The average impact force for the contusion group was 104 ± 4 kDy, and the average displacement was 582 ± 102 µm. All animals recovered successfully from the surgery. Following surgery, all animals in the contusion group exhibited bilateral hindlimb motor deficits. At the terminal experiment, upon dissection of the spinal cord, the level of injury was also confirmed in all animals by locating the T9 vertebral level and dorsal root relationship to the bruise. Five days post contusion, animals lost ~5% of initial body weight (241 ± 20 pre-injury vs 228 ± 17 at 5 d post injury).

Representative histological images of the T9 spinal segment from a spinal intact and contused animal are provided in *Figure 1*. Cross-sections from the injury epicenter in injured animals demonstrated extensive pathology of the dorsal to ventral gray matter regions and dorsal white matter tracts. There was some residual sparing of lateral and ventral white matter tracts at the injury epicenter.

## Effects of injury on cystometric bladder function and EUS EMG activity

Following the T9 contusion injury, rats acutely lost the ability to micturate spontaneously. Bladders were manually expressed 2–3 times daily for the first 3 d post injury. Urodynamic recordings were performed 5 d post contusion in injured rats and compared to spinal-intact rats. Representative urodynamic recordings are presented in *Figure 2A*. Coordinated bladder contractions and associated EUS electromyography (EMG) activity were readily demonstrated in all eight naïve animals. Cystometrogram recordings were characterized by a low baseline bladder volume during the saline-filling phase. As saline infusion reached the threshold volume, a bladder contraction occurred that was demonstrated by a pronounced rapid and brief increase in intravesical pressure and subsequently followed by voiding. Tonic EUS EMG activity was observed before the onset of voiding, followed by coordinated EUS bursting activity during the voiding phase. Distinct active and silent periods could be identified during the coordinated EUS 'bursting' phase, which coincided with intravesical pressure oscillations in the cystometrogram. By this time point, 7 out of 10 injured rats exhibited impaired but spontaneous voiding during the urodynamic recordings under anesthesia after a large threshold pressure had been achieved. In addition to cystometric deficits in the injured rats, three animals displayed uncoordinated EUS EMG activity during the voiding phase where a large increase in tonic EUS EMG activity was observed as the bladder reached its maximum capacity and distension.

Summary urodynamic data highlighting the impact of a T9 contusion on micturition are presented in *Figure 2B–I*. Baseline pressures were significantly lower in the SCI group compared to the spinal-intact group (p=0.033; *Figure 2B*). In contrast, animals with SCI had higher threshold pressures before voiding was initiated compared to the spinal-intact group (p=0.015; *Figure 2C*). Animals with SCI also had larger voided volumes (p<0.001; *Figure 2D*) and exhibited longer periods between contractions (intercontraction interval; p<0.001; *Figure 2E*). The bladder peak contraction pressures were comparable between the spinal-intact (29.6 ± 4.5 cmH$_2$O, n = 8) and SCI groups (30.3 ± 8.1 cmH$_2$O, n = 7) (p=0.83).

There was no difference in the threshold pressure for EUS EMG activation between the SCI and spinal-intact groups (p=0.95; *Figure 2F*). The duration of EUS EMG activity was significantly longer in injured rats (p<0.001; *Figure 2G*). EUS EMG activity in animals with SCI was also characterized by a significant increase in the magnitude of area under the curve (p<0.001; *Figure 2H*), and RMS$_{peak}$ EMG tended to be higher in animals with SCI compared to the intact animals (p=0.0754; *Figure 2I*).

## Effects of ampakines on bladder function

Representative urodynamic recordings following HPCD and ampakine treatment are presented in *Figure 3A*. The mean data for various urodynamic outcomes and treatments are summarized in *Supplementary file 1*. The parameters of intercontraction interval (*Figure 3B*, $F_{(4,52)}$ = 8.82; p<0.001), voided volume (*Figure 3D*, $F_{(4,52)}$ = 11.71; p<0.001), and threshold pressure (*Figure 3E*, $F_{(4,52)}$ = 19.43; p<0.001) showed a significant group × treatment interaction. Peak pressure showed a treatment effect (*Figure 3C*, p<0.001). The post hoc statistical interactions for group × treatment comparisons are summarized in *Table 1*. Levels not connected by the same letter are significantly different.

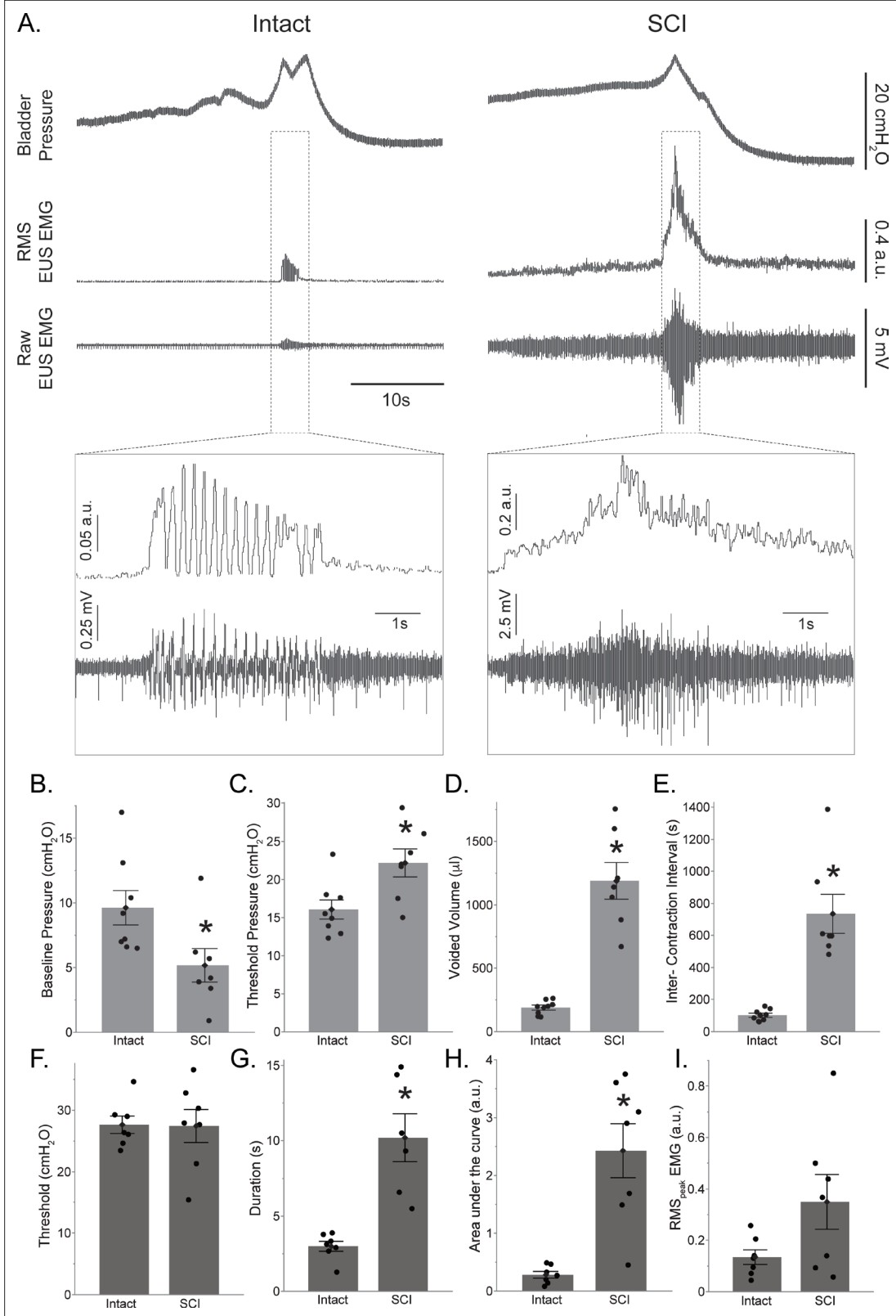

**Figure 2.** Impact of T9 contusion on cystometric bladder function and external urethral sphincter (EUS) electromyography (EMG) activity. (**A**) Example bladder pressure and EUS EMG trace of coordinated voiding in spinal intact animals (left column) and rats with spinal cord injury (SCI), 5 d following SCI (right column). Expanded traces show EUS EMG activity. Note different y-scales in expanded trace used for clarity. (**B–E**) Summary of the impact of T9 contusion injury on various cystometric outcomes. Baseline pressure was reduced in the SCI group compared to spinal intact animals. Injured

*Figure 2 continued on next page*

*Figure 2 continued*

rats had significantly higher threshold pressures, voided volume, and intercontraction intervals. (**B**) Baseline pressure (cmH$_2$O); (**C**) threshold pressure (cmH$_2$O); (**D**) voided volume (μl); (**E**) intercontraction interval (s). (**F–I**) Summary of the impact of T9 contusion injury on EUS EMG activity. Injured rats had a significantly higher duration, area under the curve, and RMS$_{peak}$ EMG. (**F**) Threshold (cmH$_2$O); (**G**) duration (s); (**H**) area under the curve (arbitrary unit, a.u.); (**I**) RMS$_{peak}$ EMG (a.u.). *p<0.05. Data are presented as bar plots, with all individual data points corresponding to individual animal means. Group means are represented with a diamond, with error bars depicting ±SE.

Spinal-intact rats demonstrated no effect of HPCD or ampakine treatment on the intercontraction interval, or voided volume. However, 10 and 15 mg/kg ampakine treatment significantly reduced peak pressures in spinal-intact rats (28.3 ± 6.7 cmH$_2$O with HPCD vs 23.8 ± 4.3 cmH$_2$O with 10 mg/kg ampakine and 23.5 ± 3.9 cmH$_2$O with 15 mg/kg ampakine, n = 8; treatment effect; p<0.001, *Figure 3C*). Ampakine treatment with doses of 10 and 15 mg/kg caused an ~25% reduction in the threshold pressure in these rats (treatment effect, p<0.001, *Figure 3E*).

In rats with SCI, infusion of the HPCD vehicle had no discernible impact on intercontraction interval, peak pressure, voided volume, and threshold pressure. In sharp contrast, the 10 and 15 mg/kg ampakine doses caused an ~55% decrease in the intercontraction interval (treatment effect; p<0.001, *Figure 3B*). Similarly, to spinal-intact rats, peak pressure decreased in the 10 and 15 mg/kg ampakine-treated rats with SCI (27 ± 5 cmH$_2$O with HPCD vs 20.9 ± 7.8 cmH$_2$O with 10 mg/kg ampakine and 20.7 ± 8.4 cmH$_2$O with 15 mg/kg ampakine, n = 7) (treatment effect; p<0.001; *Figure 3C*). Compared to the baseline and HPCD treatment, each of the three ampakine doses also resulted in a reduction in voided volume (~32–58%, treatment effect, p<0.001, *Figure 3D*), likely owing to the shorter intercontraction intervals and higher frequency of voiding. Similarly, 5, 10, and 15 mg/kg ampakine doses caused a reduction (~50–58%, treatment effect, p<0.001, *Figure 3E*) in threshold pressure compared to the baseline and HPCD pressure.

## Effects of ampakine on EUS EMG activity

Representative EUS EMG traces following HPCD and ampakine treatment are presented in *Figure 4A and B*. The mean data for various EUS EMG parameters at baseline and following treatments are summarized in *Supplementary file 2*. The EUS burst duration was considerably longer in injured rats compared to spinal-intact rats (*Figure 4C*, p<0.001). Neither HPCD nor ampakine treatment had any effect on EMG burst duration in injured or spinal-intact rats (treatment effect; p=0.52). However, there was a robust treatment effect on the threshold bladder pressure for evoking EUS EMG activity (*Figure 4D*, p<0.001). In both injured and spinal-intact rats, ampakine treatment caused EUS EMG bursting to be evoked at much lower bladder pressures compared to baseline. This effect was also dose-dependent as the 10 and 15 mg/kg doses elicited a larger drop in threshold levels. Ampakine or HPCD treatment had no effect on the magnitude of area under the curve (*Figure 4E*, p=0.22) or RMS$_{peak}$ EMG (*Figure 4F*, p=0.20).

## Effects of ampakine on non-voiding rats 5 d following SCI

A small cohort of animals (3 of our 10 rats) could not establish a regular cystometric voiding pattern, as evidenced by many non-voiding contractions and slow leaking once the bladder was full (*Figure 5A*). Manual expression of the bladder had to be performed in order to void the bladder during cystometry. This group of animals was assessed separately from the animals displaying spontaneous voiding since the parameters of intercontraction interval, base pressure, threshold pressure, or peak pressure could not be measured during baseline conditions in the non-voiding animals. Ampakine treatment (10 mg/kg) was successful in reducing the number of non-voiding and partial contractions in all three rats and increasing the frequency of complete voiding contractions (*Figure 5B–D*). Interestingly, we observed the reappearance of EUS EMG bursting events in rats with SCI with inefficient coordinated voiding by administration of CX1739 (*Figure 4B*) and clear coordinated bladder contraction with EUS EMG bursting activity with CX1739 in all three rats with SCI where there was complete loss of coordination and failed to produce voiding (*Figure 5E and F*). These results suggest that ampakines potentiate micturition circuits in rats with SCI at a level that was able to produce a coordinated void.

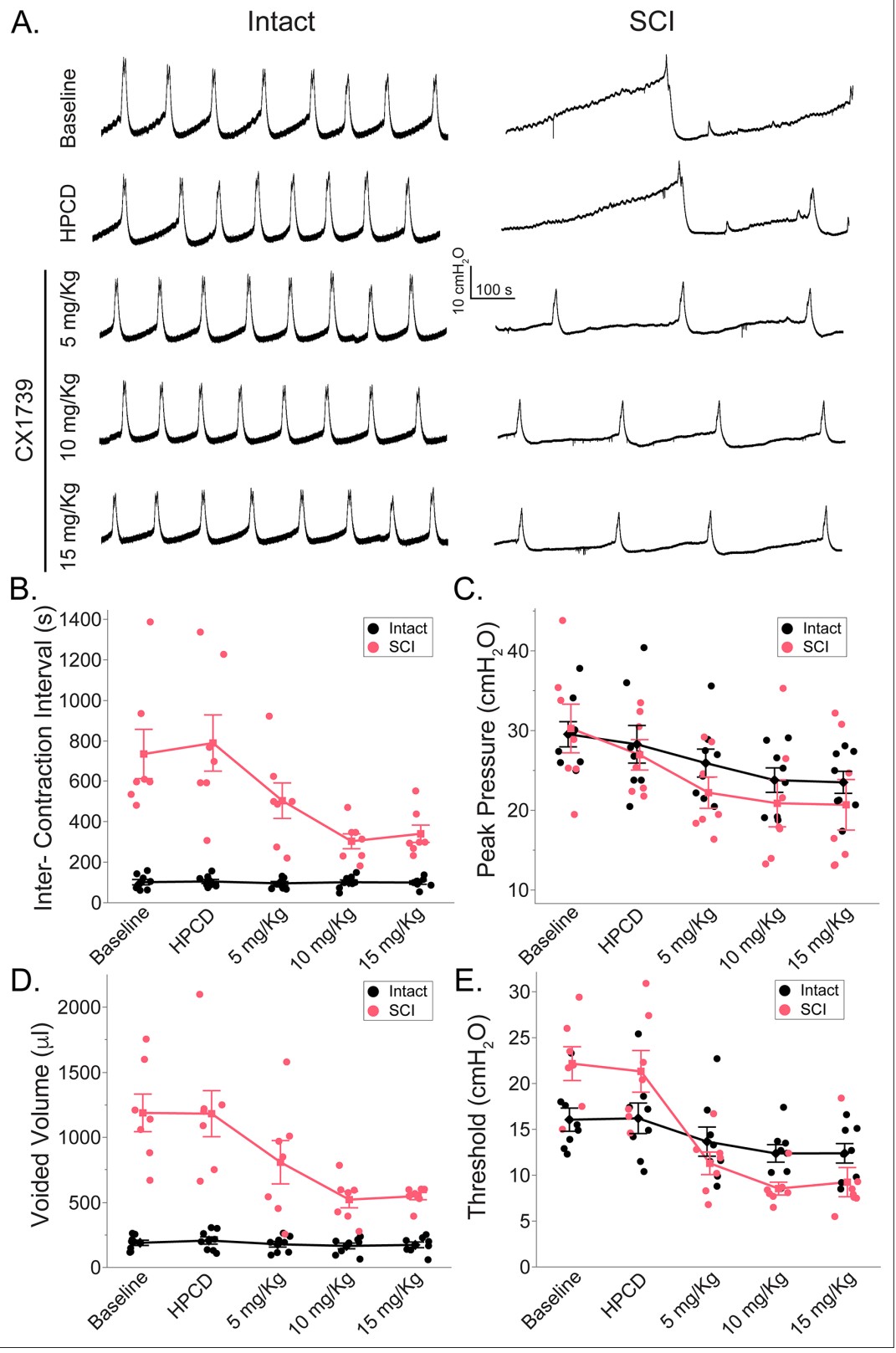

**Figure 3.** Impact of ampakine treatment on cystometric bladder function. (**A**) Example trace of bladder cystometry in spinal intact animals (left column) and rats with spinal cord injury (SCI) 5 d following SCI (right column) and following 2-hydroxypropyl-beta-cyclodextrin (HPCD) or ampakine infusion (5, 10, 15 mg/kg). (**B–E**) Summary of the impact of ampakine treatment on cystometric outcomes in spinal intact (n = 8) rats and rats with SCI (n = 7).

*Figure 3 continued on next page*

*Figure 3 continued*

Ampakine treatment significantly reduced the intercontraction interval, voided volume, and threshold pressure in injured rats. HPCD did not alter cystometry parameters compared to baseline. Ampakine treatment caused a decrease in threshold and peak pressure but did not affect intercontraction interval or voided volume in intact rats. (**B**) Intercontraction interval (s); (**C**) peak pressure (cmH$_2$O); (**D**) voided volume (μl); (**E**) threshold pressure (cmH$_2$O). Data are presented as line plots, with all individual data points corresponding to an individual animal means. Group means are represented with a diamond, with error bars depicting ±SE.

## Discussion

We demonstrate that systemic (i.v.) delivery of a positive allosteric modulator of AMPA receptors, ampakine CX1739, rapidly improves micturition reflexes in subacute spinally injured rats. In particular, acute ampakine treatment reduced SCI-induced deficits in the intercontraction interval, threshold pressure, and peak pressure. A decrease in voided volume was also observed with the ampakine treatment, which would reduce the potential for kidney damage in rats with SCI. Ampakine treatment reduced the pressure threshold of EUS bursting initiation during the micturition cycle. Ampakines have been safely administered in human clinical trials for other purposes (*Boyle et al., 2012*; *Wesensten et al., 2007*), and accordingly, our results suggest that ampakine pharmacotherapy may be a new strategy to support the recovery of bladder function following acute spinal cord injuries.

### Therapeutic impact of ampakines

Ampakines are designed to enhance AMPA-mediated glutamatergic neurotransmission (*Arai and Kessler, 2007*; *Lynch, 2006*). In vitro studies confirm that ampakines are not AMPA receptor agonists but are allosteric modulators of their activity (*Arai et al., 1996*; *Arai et al., 2002*; *Arai et al., 2004*). Ampakines do not impact NMDA or kainite receptors directly (*Lynch, 2006*). Importantly, ampakines exist in two distinct classes: high- and low-impact ampakines. High-impact ampakines act mainly by changing AMPA receptor desensitization or changing the affinity of an agonist binding to the receptor (*Arai et al., 2002*). There have been concerns with the use of high-impact ampakines, which can cause hyper-excitability of neuronal circuits and might be limited by a narrow therapeutic window. The mode of action of low-impact ampakines is primarily via changing the rate at which channels open, resulting in a small increase in current amplitude without prolonging channel deactivation (*Arai et al., 2002*). The positive effects of low-impact ampakines in rodent models (*Arai and Kessler, 2007*; *Lynch, 2006*; *Lynch and Gall, 2006*), without the convulsant effects (*Shaffer et al., 2013*) of high-impact ampakines,

**Table 1.** Statistical post hoc comparisons for cystometric outcomes with significant treatment and group interactions in spontaneously voiding intact (n = 8) rats and rats with SCI (n = 7) following ampakine treatment.

| Group | Treatment | Threshold | Intercontraction interval | Voided volume |
|-------|-----------|-----------|---------------------------|---------------|
| Intact | Baseline | A, B | D | C |
| | HPCD | A, B, C | D | C |
| | 5 mg/kg | B, C, D | D | C |
| | 10 mg/kg | D | D | C |
| | 15 mg/kg | D | D | C |
| SCI | Baseline | A | A, B | A |
| | HPCD | A | A | A |
| | 5 mg/kg | B, C, D | B, C | B |
| | 10 mg/kg | C, D | C, D | B, C |
| | 15 mg/kg | D | C, D | B, C |

Levels not connected by the same letter are significantly different.
HPCD, 2-hydroxypropyl-beta-cyclodextrin; SCI, spinal cord injury.

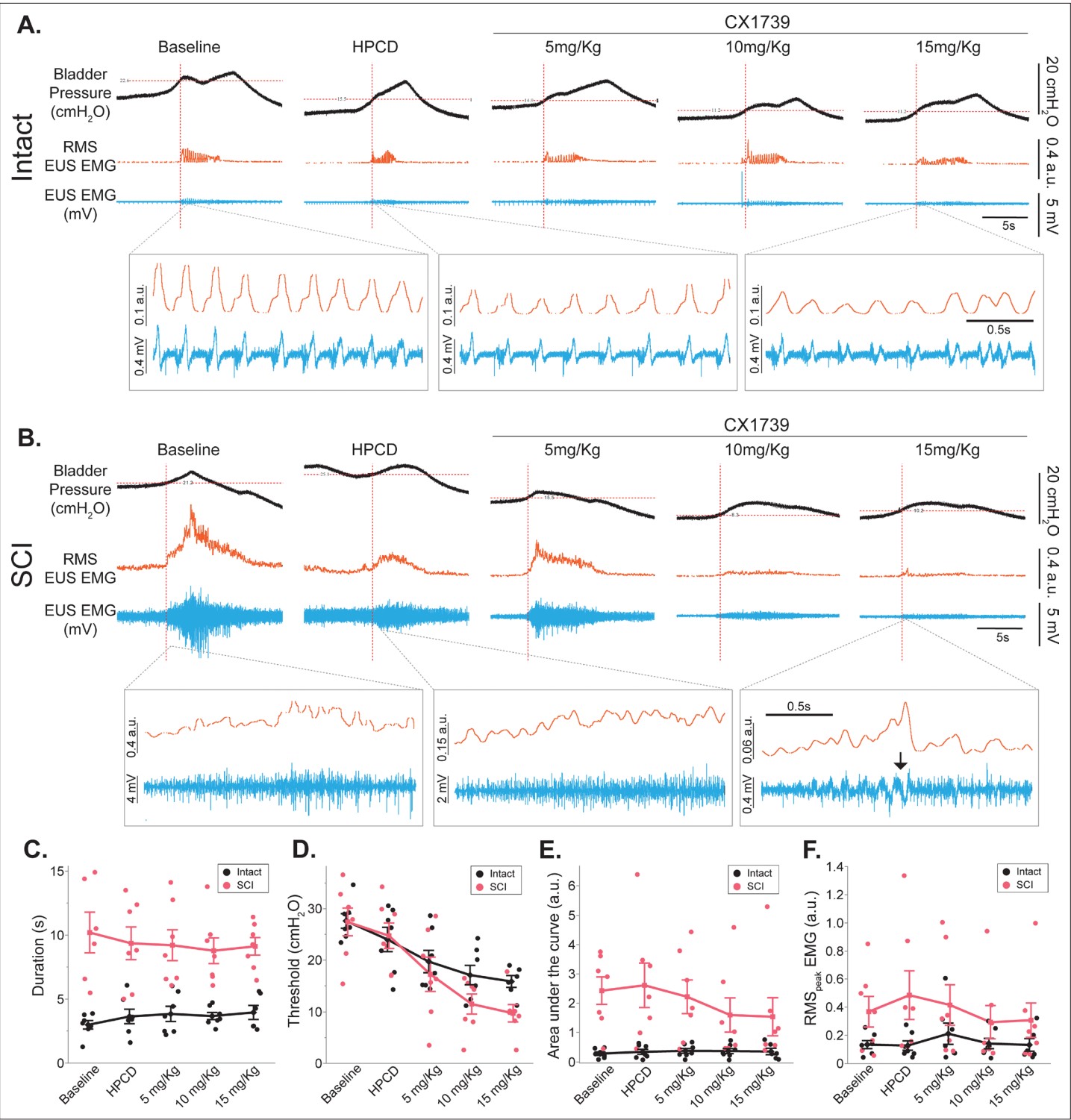

**Figure 4.** Impact of ampakine treatment on external urethral sphincter (EUS) electromyography (EMG) activity. (**A, B**) Example bladder pressure trace, raw EUS EMG trace (blue), and RMS EUS EMG (orange) activity during spontaneous voiding in spinal intact animals and rats with spinal cord injury (SCI) 5 d following SCI following 2-hydroxypropyl-beta-cyclodextrin (HPCD) or ampakine infusion (5, 10, 15 mg/kg). Note the re-emergence of coordinated EUS EMG activity following 15 mg/kg ampakine treatment in injured rats (black arrow). (**C–F**) Summary of the impact of ampakine treatment on EUS EMG activity in spinal intact rats (n = 8) and rats with SCI (n = 7). Ampakine treatment reduced the threshold pressure in spinal intact rats and rats with SCI. HPCD had no impact on any outcomes. (**C**) Duration (s); (**D**) threshold (cmH$_2$O; defined as the bladder pressure at which EUS EMG burst was evoked [intersection of red dotted lines]); (**E**) area under the curve (arbitrary unit, a.u.); (**F**) RMS$_{peak}$ EMG (a.u.). Data are presented as line plots, with all individual data points corresponding to individual animal means. Group means are represented by a diamond, with error bars depicting ±SE.

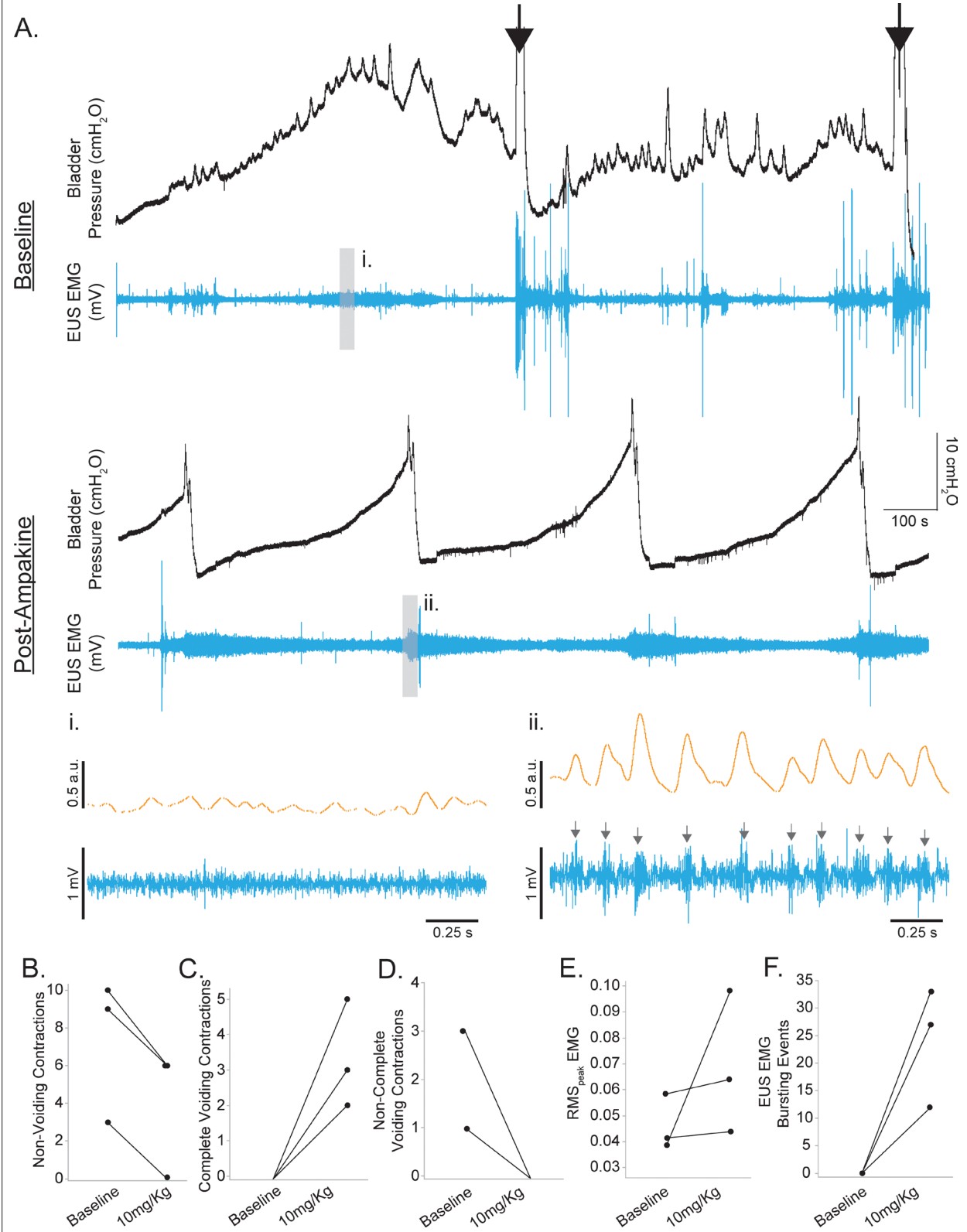

**Figure 5.** Ampakine rescue coordinated voiding in non-voiding rats following spinal cord injury (SCI). (**A**) Example bladder pressure and external urethral sphincter (EUS) electromyography (EMG) trace of non-coordinated voiding 5 d following SCI (baseline). Black arrows indicate manual expression/emptying of the bladder. The second trace (post-ampakine) indicates the same rat following 10 mg/kg CX1739 with voiding re-established. Expanded EUS EMG trace in the middle panels shows the lack of coordinated bursting during baseline recording of rats with SCI (i). Following

*Figure 5 continued on next page*

*Figure 5 continued*

ampakine treatment, coordinated EUS bursting is restored (gray arrows; ii). (**B–F**) Summary of the impact of 10 mg/kg ampakine treatment on cystometry and EUS EMG outcomes in rats with SCI displaying non-coordinated voiding (n = 3). Ampakine treatment reduced non-voiding and partial voiding contractions in all three rats and established coordinated voiding. No rat exhibited coordinated EUS bursting during the voiding phase at baseline. Ampakine treatment re-established coordinated EUS bursting in all three rats. (**B**) Non-voiding contractions; (**C**) complete voiding contractions; (**D**) non-complete voiding contractions; (**E**) RMS$_{peak}$ EMG; (**F**) EUS EMG bursting events. Data are presented as individual data points corresponding to individual animal means.

make them a better therapeutic candidate. For this reason, our group's current and prior studies have focused on low-dose, low-impact ampakines following SCI.

We have recently demonstrated that low-impact ampakines can enhance respiratory drive following SCI (*Rana et al., 2021*; *Wollman et al., 2020*). High- and mid-cervical SCIs result in respiratory compromise due to disrupted glutamatergic pathways that activate motor neurons controlling the diaphragm muscle. AMPA receptors play a large role in mediating this respiratory drive (*Thakre et al., 2023*). In acute and subacute stages following a high-cervical SCI, systemic delivery of a low-dose ampakines (CX1739 and CX717, 5 mg/kg) was sufficient to enhance diaphragm EMG activity and overall ventilation in freely behaving rats. Further, systemic delivery of CX1739 or CX717 at 5 mg/kg resulted in no detectable off-target effects, such as changes in heart rate, blood pressure, or overall animal activity. In addition to its efficacy in improving respiratory function in rodent models of SCI, CX1739 was safely administered with no adverse events as part of phase 2 clinical trials to alleviate opioid-induced respiratory depression (*RespireRx University D, 2016*).

While pharmacokinetic studies were not conducted in the present study, the mean plasma half-life of a single CX1739 intravenous bolus in Sprague–Dawley rats is 1.25 ± 0.03 hr, with a T$_{max}$ (time to maximum plasma) concentration of 30 min (information provided through personal communication with RespireRx). The mean range of CX1739 half-life in humans has been reported to be in the range of 6.6–9.3 hr. In our studies, we continued to observe an effect of ampakines for the duration of 45 min post-drug administration. Although the current acute preparations were constrained by time to reduce variability in cystometry recordings and keep the total duration of the preparation under 6 hr, future studies will aim to test the long-term impact of an acute and optimal ampakine dose in awake rats, a critical step along the translational pathway.

Lastly, in the present study, only female rats were studied due to the lower incidence of bladder complications following a thoracic contusion injury. Future studies will need to be completed in both male and female rats to compare potential sex differences in ampakine effects. While we have not observed any sex effects of ampakines in our respiratory studies (*Rana et al., 2021*), there are known differences in recovery and injury manifestation between male and females that need to be further studied (*Anderson et al., 2023*; *Myers et al., 2023*).

## Role of AMPA receptors in micturition circuits

Glutamate signaling through the AMPA-type receptors plays a critical role in several micturition circuits under normal conditions and after SCI (*de Groat et al., 2015*; *Fowler et al., 2008*). In general, glutamate is excitatory in bladder circuits, and AMPA receptors play a major role in mediating glutamatergic drive. Indeed, AMPA receptor antagonists can decrease bladder contraction frequency and reduce EUS EMG activity (*Yoshiyama et al., 1997*). These actions occur via the descending input from the pontine micturition center to the pre-ganglionic detrusor and urethral sphincter motor neurons. AMPA receptors are also expressed in bladder sensory circuits, where inhibition of AMPA receptors reduces sensory input to the brain stem (*Kakizaki et al., 1998*). They are presumably present on the first-order spinal cord neurons responding to glutamate released by sensory neurons, but this has yet to be proven definitively.

Based on data from the current study, the increased incidence of coordinated bladder voiding in injured rats following systemic ampakine administration would support a centrally mediated pathway involving both the sensory and motor systems (*Kakizaki et al., 1998*; *Matsumoto et al., 1995a*; *Matsumoto et al., 1995b*). The ascending and descending pathways express different AMPA receptor subunits (*Shibata et al., 1999*). Specifically, dorsal laminae in the L6-S1 rat spinal cord constitute a high abundance of GluR1 and GluR2 subunits (*Shibata et al., 1999*). Whereas neurons located in the ventral horn have a high abundance of GluR3 and GluR4 subunits. In pontine micturition centers, high

levels of GluR1 and GluR2 subunits have been observed (*Shibata et al., 1999*). Positive allosteric modulators have been shown to bind at the interface between two adjacent subunits at the D1 lobe of the ligand-binding domain (*Jin et al., 2005*). Alternative splicing of amino acids at the ligand-binding domain of various receptor subunits can change the domain interface, thereby changing the interaction of allosteric modulators and how they impact channel kinetics and their sensitivity (*Golubeva et al., 2022*). Based on current data, we presume that ampakines act similarly on various AMPA receptor subunits. However, future studies will need to determine the location of ampakines' action to modulate bladder function following incomplete SCI.

Unlike previous studies using AMPA receptor antagonists, we did not observe any statistically significant effects of CX1739 on bladder contraction or EUS EMG activity in spinal-intact rats. Previous studies in naïve rats have shown that AMPA receptor antagonism with LY215490 inhibits detrusor and EUS activity (*Yoshiyama et al., 1997*). It is important to note that in our study we used CX1739, which is not an AMPA receptor agonist but an allosteric modulator of the receptor that can enhance endogenous signaling. The AMPA antagonist used in the previous study, LY215490, antagonizes glutamate to bind with postsynaptic AMPA receptors and, thereby, completely inhibits the glutamatergic synaptic transmission of micturition circuits, resulting in complete inhibition of bladder and EUS activity. In comparison, CX1739 is an allosteric modulator of AMPA receptors and does not directly activate the receptor. Thus, our results indicate that allosterically modulating the receptor with a low-impact ampakine does not enhance normal glutamatergic synaptic transmission in spinal-intact rats, while it can enhance activity following SCI. Future research is needed to determine why this is the case, but it could be due to downregulation of AMPA receptors (*Grossman et al., 1999*) and increased release of glutamate in SCI models (*Demediuk et al., 1989*; *Liu et al., 1991*; *Panter et al., 1990*).

## Role of AMPA receptors in micturition circuits after SCI

After SCI, the bladder is initially in an 'areflexive state' and micturition is impaired. As spinal circuits undergo post-SCI neuroplastic changes, the bladder transitions to a neurogenic detrusor overactivity (*de Groat et al., 2015*; *de Groat and Yoshimura, 2006*). This results in a hyper-reflexive state and causes sphincter dyssynergia, reflex incontinence, and increased residual urine in the bladder following a void. It is well established that glutamatergic signaling and expression of AMPA receptors are profoundly altered after SCI (*Grossman et al., 2001*; *Grossman et al., 1999*; *Pikov and Wrathall, 2002*; *Rana et al., 2022*; *Yoshiyama et al., 1999*). In the thoracic spinal cord, there is a decrease in the expression of AMPA receptor subunits acutely (e.g., 24 hr) after thoracic SCI (*Grossman et al., 1999*). However, AMPA receptor expression increases at more chronic post-SCI time points (e.g., 1 mo) following the injury (*Grossman et al., 1999*). Similar dynamic changes in AMPA receptors have been described in other spinal segments after SCI (*Rana et al., 2022*). An upregulation of AMPA receptors in the chronic stages of injury may, at least in part, be responsible for the hyperactive bladder phenotype that can occur after chronic SCI (*Pikov and Wrathall, 2001*; *Pikov and Wrathall, 2002*; *Yoshiyama et al., 1999*). It is unclear why these receptors are upregulated, although this could occur due to homeostatic upregulation (*Turrigiano, 2012*) secondary to decreased activity in denervated motoneurons due to the SCI-induced interruption of descending tracts.

Our data indicate that ampakine-mediated modulation of AMPA receptors can rescue the areflexive bladder phenotype 5 d after SCI. At this acute stage post-contusion injury, glutamatergic neurotransmission is partially impaired due to disrupted descending spinal pathways. Specifically, excitatory glutamatergic pathways originating from the pontine micturition centers would be partially disrupted following a T9 contusion injury. Indeed, histological assessment of the spinal cord tissue (*Figure 1*) shows partial disruption of white matter regions that would contain descending lateral corticospinal tracts from the pontine micturition centers (*Fowler et al., 2008*). In addition, ascending sensory tracts contributing to the sensation of bladder filling would also be disrupted. Thus, in this state of reduced neural drive, ampakines are likely potentiating residual drive to activate the glutamatergic pathways responsible for the areflexive bladder. These effects of ampakines in the acute phase of injury are particularly exciting. In the chronic phase of injury, it is well documented that detrusor overactivity resulting from hyperactivity glutamatergic circuits tends to develop at the 3- to 4-week post-injury time points (*Sartori et al., 2022*). While ampakines appear to alleviate bladder dysfunction at this acute time point post injury, it is likely that there is a time-dependent efficacy of ampakine therapy that changes from acute to chronic stages of injury, and these aspects remain to be studied.

Globally more than 500,000 people experience an SCI every year, of which 70–84% of patients showed neurogenic bladder dysfunction (*de Groat et al., 1990*; *Kumar et al., 2018*). Current therapeutic options for improving bladder dysfunction in SCI patients are not effective and rely primarily on intermittent catheterizations to prevent bladder overfilling and kidney damage. These approaches may further lead to many bladder and kidney function problems (*Xiang et al., 2023*). In the present study, we show that CX1739 acutely improves hyporeflexive deficits in bladder function in rats following SCI, as measured by cystometry. Cystometry is an excellent tool to evaluate bladder filling and emptying conditions for drug dose responses as data can be collected from multiple voids during a relatively short drug activation period. This technique does have its limitations as it is conducted under anesthetized conditions and using non-physiological filling rates. A critical step in future studies, before translation into the clinical population, will be to evaluate overall bladder voiding in freely moving animals and how bladder function changes in the acute to chronic stages post-injury in ampakine-treated rats.

## Conclusions

We conclude that intravenous delivery of a low-impact ampakine, CX1739, can improve bladder voiding after thoracic SCI. In rats with impaired bladder voiding due to SCI, CX1739 treatment increased the frequency of coordinated voiding and promoted coordinated EUS EMG activity. Importantly, low-impact ampakines have been safely administered to humans in clinical trials for other indications (*Boyle et al., 2012*; *Wesensten et al., 2007*). Accordingly, we suggest that ampakine pharmacotherapy may represent a viable strategy to improve acute hyporeflexive bladder function in persons with SCI.

## Methods

### Experimental animals

A total of 18 adult (12 weeks old) female Sprague–Dawley rats (Hsd:Sprague Dawley SD, Envigo Indianapolis, IN) were used in the study. Most previous rat studies on SCI and bladder function used females since there are considerably fewer postoperative complications and due to the ease of postoperative manual voiding when the rats are unable to self-void (*Lin et al., 2016*; *Mitsui et al., 2014*; *Pikov and Wrathall, 2001*). Our initial preliminary experiments included male and female rats; however, only female rats recovered cystometric voiding 5 d after injury. For these reasons, female rats were chosen for this study.

All procedures were approved by the Institutional Animal Care and Use Committee at the University of Florida and are in accordance with the National Institutes of Health Guidelines (protocol # 202107438). Animals were housed individually in cages under a 12 hr light/dark cycle with ad libitum access to food and water.

### T9 contusion injury

A cohort of rats in this study received a midline T9 contusion injury (n = 10). Depth of ketamine (90 mg/kg) and xylazine (10 mg/kg) anesthesia was confirmed at the onset of any procedure using a lack of hindlimb withdrawal or whisker twitch following a toe pinch. An adequate level of anesthesia was continuously verified during the surgical procedure, and the respiratory rate was measured every 15 min. The animal was re-dosed with a 1/3 dose of the initial ketamine/xylazine dose if needed. Body temperature was continuously monitored using a rectal temperature probe and maintained between 37 and 38°C using a water-recirculating heating pad. Under sterile conditions, a dorsal incision was made from the T5–T11 region of the spine. Dorsal paravertebral muscles between T6–T10 were incised and retracted. The posterior portion of thoracic vertebrae was exposed at T8–T10, and a laminectomy was performed at T9 while preserving the facet joints and leaving the dura intact. Rats were suspended by clamps secured laterally at the T8 and T10 vertebrae. A 2.5 mm impactor tip was aligned at the midline. Subsequently, rats were subjected to a single contusion injury. A desired force of 100 kDy with 0 s dwell time was delivered using the Infinite Horizon Impactor (Precision Systems and Instrumentation, Lexington, KY). Animals were de-clamped and the overlying muscles were sutured with sterile 4-0 webcryl. The overlying skin was closed using 9 mm wound clips. Animals were maintained on a heating pad until alert and awake. Animals received one dose of the extended-release buprenorphine SR LAB (1 mg/kg, ZooPharm), followed by carprofen (5 mg/kg, q.d.), baytril

(5 mg/kg, q.d.), and lactated Ringer's solution (10 ml/d, q.d.) for the initial 48 hr post injury. Animals were monitored daily for signs of distress, dehydration, and weight loss, with appropriate veterinary care given as needed. No animals needed to be excluded from the study based on predetermined exclusionary factors of bone-hit or slip tip during contusion, >10% difference in actual delivered force, limb autophagia, or weight loss >20% of a pre-injury time point.

## In vivo bladder cystometry recordings

Five days following SCI or in intact rats, a 24 G i.v. catheter (SR#FF2419, Terumo Corporation) was placed on the distal end of the tail vein on the day of recording under 2% isoflurane. Rats were slowly infused with urethane (0.84 g/kg; 102452447, Sigma; diluted in saline at 150 mg/ml) through the tail vein. Later 0.12–0.24 g/kg of urethane was given if required during cystometry, depending on the animal's response. An injection of saline was given subcutaneously (1 ml/100 g of rat) to keep the rat hydrated during the recording time of cystometry under urethane. The abdominal area and the area around the base of the tail were shaved. The rat was then transferred to a closed-loop heating pad to control the body temperature at 37°C (2221962P, CWE, Inc). The rat was placed under 1–2% isoflurane (J121008, Akron, Inc) through a nose cone. An incision was made in the abdominal wall, and a purse-string suture was made using a 5-0 nylon suture (07-809-8813, Patterson Veterinary) around the apex of the bladder. The apex of the bladder inside the purse-string suture was cut, and a flared catheter (BB31695-PE/3, Scientific Commodities, Inc) was inserted into the bladder lumen. The purse-string suture was secured tightly around the catheter, and the bladder was checked for leaks with a brief saline infusion. The polyethylene catheter was connected with a syringe pump (GT1976 Genie Touch, Kent Scientific Corporation) to infuse the saline into the bladder at 6 ml per hour. The intraluminal pressure measurements were acquired by using an inline pressure transducer (503067, World Precision Instruments) connected to a Transbridge amplifier (TBM4-D, World Precision Instruments) and processed at a 1000 Hz sampling rate using the Micro 1401 (Cambridge Electronic Design) data acquisition system and Spike2 Software (V10; Cambridge Electronic Design). The muscle of the abdominal cavity and skin incisions were sutured with nylon suture, and the isoflurane was then lowered to zero.

Next, two Teflon-insulated silver wires (570742, A-M Systems) were placed into the EUS muscle (5 mm of bare wire exposed) to measure sphincter EMG. Two 25 G needles containing the EMG wires were inserted into the skin and muscle at a distance of 3–5 mm from the urethral opening, and the other end of the wire was connected with the pre-amplifier (HZP, Grass, Astro-Med. Inc) and then amplifier (RPS107E, Grass, Astro-Med Inc). A third wire was inserted into the skin of the abdominal area of the body by using a 18 G needle and connected to the pre-amplifier as a ground. The EUS EMG data from the amplifier was processed at 20,000 Hz using the Micro 1401 data acquisition system and Spike 2 software (*Figure 6A*). Then, 30 min after stopping isoflurane, we started the saline infusion and cystometric pressure recording. The voids were collected in a weigh boat by placing the weigh boat under the urethral opening to collect the voided volume. These volumes were weighed on a balance and recorded after each cystometric void.

## Chemical and pharmacological agents

Ampakine CX1739 was provided by RespireRx. The drug was diluted in HPCD (Cat# H107-100G, Sigma) solution (10%) at a soluble concentration of 5 mg/ml. Aliquots were stored at −20°C for up to 6 mo. Aliquots were thawed to room temperature on the day of each experiment. CX1739 was infused at concentrations of 5, 10, and 15 mg/kg intravenously slowly (~2 min). We then assessed the effects of those doses of CX1739 over the next 45 min. Cystometry parameters were recorded as baseline, vehicle, 5 mg/kg, 10 mg/kg, and 15 mg/kg CX1739 in the same order (*Figure 6B*). Pharmacokinetic parameters for CX1739 in Sprague–Dawley rats have previously been determined following an intravenous administration of CX1739. The mean plasma half-life of CX1739 was 1.25 ± 0.03 hr, with a $T_{max}$ of 30 min (information provided through personal communication with RespireRx). Although the 45 min interval between doses would not be within the time frame of *complete* post-administration clearance of the first CX1739 dose from the system, the plasma levels would be predicted to be considerably lower by 45 min post administration. A limitation of terminal cystometry preparations is that the duration which an experiment in an anesthetized rat can be sustained is limited. In our experience, recordings beyond 6 hr will dramatically increase variability in the data. The 45 min window allowed for the procedure to remain under ~6 hr. Further, in our published studies investigating the

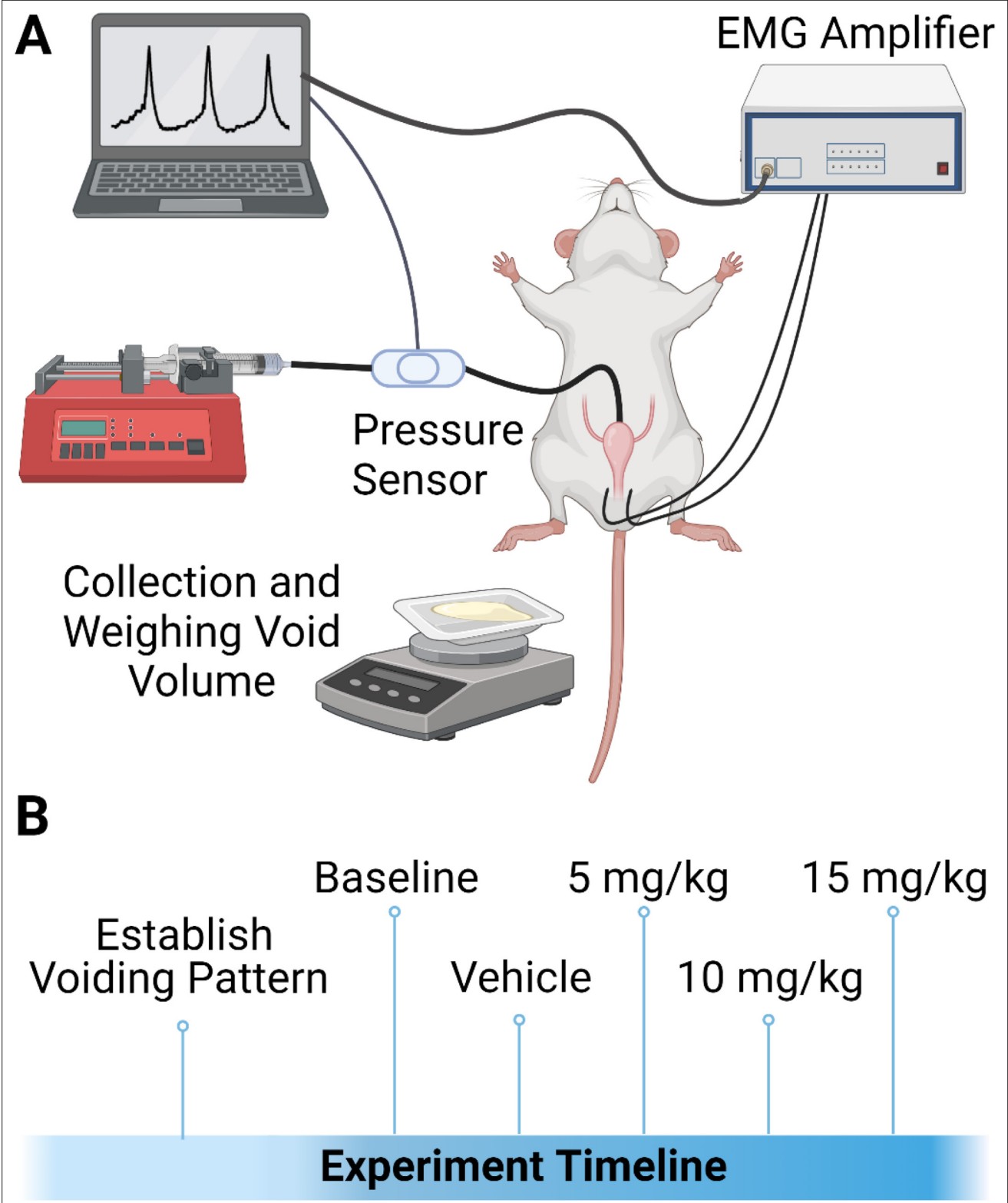

**Figure 6.** Experimental setup and design. (**A**) Schematic of experimental urodynamic recordings in anesthetized rats. Volume of voided urine was also collected. (**B**) Experimental timeline of studies. Animals received a unilateral T9 contusion, and urodynamic recordings were conducted at 5 d post contusion. During the recording, a baseline was established for urodynamic output. Animals received vehicle 2-hydroxypropyl-beta-cyclodextrin (HPCD), and consecutive doses of ampakines CX1739 at doses of 5, 10, and 15 mg/kg. Each drug dose was spaced apart by 45 min.

impact of ampakines on breathing in rats following an SCI, the acute impact of intravenous ampakine administration wanes after approximately 30 min (*Rana et al., 2021*) Thus, the dosing schedule was based on three factors: (1) the drug half-life, (2) the duration of which the experimental preparation was viable, and (3) the data from the respiratory response of ampakines after SCI.

## Immunohistochemistry and imaging

Following the cystometry experiments, animals were injected intraperitoneally with ketamine/xylazine cocktail (0.05 ml/100 g) and transcardially perfused with 4% paraformaldehyde in 0.1 M phosphate-buffered saline (pH 7.4). T8–T10 spinal segments were resected, post-fixed for 24 hr in 4% paraformaldehyde, and cryopreserved in 30% sucrose in 0.1 M PBS (pH 7.4) for 3 d at 4°C. A 5 mm spinal cord segment centered at the injury epicenter was embedded in OCT and subsequently sectioned at 20 μm thickness in the transverse plane. Rehydrated sections were stained with 0.1% cresyl violet in glacial acetic acid. After 3 min, slides were rinsed in water and dehydrated through graded alcohol steps followed by xylene. Tissue sections were imaged and stitched using a ×10 objective on a Keyence microscope (BZ-X700, Keyence Corporation of America, Itasca, IL).

## Data analysis

Animals that did not develop a regular voiding pattern were not included in the group analysis as cystometric properties could not be measured. Of 10 rats with SCI, 3 did not develop a normal pattern, all intact rats exhibited a regular pattern (8 out of 8). These data were analyzed separately by making comparisons within animals (*Figure 5*). Cystometric data (intercontraction interval, threshold pressure, peak pressure, and baseline pressure as defined by *Andersson et al., 2011*) was first analyzed using cystometry analyzer software (*Heitmeier and Samineni, 2022*; https://github.com/Samineni-Lab/cystometry_analyzer). Threshold pressures were verified manually to ensure accuracy. The cystometric voided saline was collected in a weigh boat and weighed on a balance to determine the volume of the voids. The total calculated volume of voids over a period of 30 min after the application of HPCD/CX1739 was divided by the total number of voids to determine the average voided volume. EUS EMG data was analyzed on Spike2 software (CED). The EMG signal was rectified, moving-median filtered using a 50 ms time constant, and finally smoothed using a 50 ms time constant to obtain the RMS EMG signal. The peak RMS EMG signal averaged over the duration of the burst to obtain $RMS_{peak}$ EMG. The threshold activation pressure for EUS EMG was defined as the bladder pressure at which EUS EMG burst was evoked (intersection of red dotted lines in *Figure 4A*).

## Statistical analysis

All statistical evaluations were performed using JMP statistical software (version 14.0, SAS Institute Inc, Cary, NC). Power calculations for each cohort of animals in the study were determined before the initiation of the study and based on pilot experiments not included in this article. The study statistical design (n = 8 rats/group) was powered to consider the expected standard deviation in cystometrogram threshold pressure at 5 d post-contusion (22%) and in order to detect a 25% difference at a power of 0.8 and an α of 0.05. Using the mixed linear model, statistical significance was established at the 0.05 level and adjusted for any violation of the assumption of sphericity in repeated measures using the Greenhouse–Geisser correction. Group (intact or injured) and treatment (baseline, HPCD, CX1739 5 mg/kg, CX1739 10 mg/kg, and CX1739 15 mg/kg) were included as model variables using animals as a random effect. When appropriate, post hoc analyses were conducted using Tukey–Kramer honestly significant difference. Normality of the distribution was assessed using the Shapiro–Wilk test within each animal. Outliers were identified for each animal using an outlier box plot. No data were excluded from the analysis to highlight the responders versus the non-responders in each treatment group. Each data point on the graphs represents mean ± SEM. Figures were designed and produced using JMP and Adobe Illustrator.

## Acknowledgements

We gratefully acknowledge Dr. Arnold Lippa and RespireRx for supplying the ampakines used in this work. We also thank the Animal Care Services (ACS) staff of the University of Florida for providing oversight for the care and well-being of all animals used in this study. *Figure 1* was created using Biorender.com. This work was supported by NIH 1R01HL139708-01A1 (DDF), SCIRTS Craig H Neilsen

Foundation (SR), a grant from the Rita Allen Foundation Scholars Program Fund, a component fund of the Community Foundation of New Jersey (AM), the NIH NIBIB Trailblazer award (R21 EB031249), the 2022 Urology Care Foundation Research Scholar Award Program, and the Indian American Urological Association Sakti Das, MD Awards (FA).

## Additional information

### Funding

| Funder | Grant reference number | Author |
|---|---|---|
| National Institutes of Health | 1R01HL139708-01A1 | David D Fuller |
| National Institutes of Health | R21 EB031249 | Aaron D Mickle |
| Craig H. Neilsen Foundation | SCIRTS | Sabhya Rana |
| Rita Allen Foundation | Scholars Program Fund | Aaron D Mickle |
| Urology Care Foundation | 2022 Research Scholar Award Program | Firoj Alom |
| Indian American Urological Association | Sakti Das, MD Award | Firoj Alom |

The funders had no role in study design, data collection and interpretation, or the decision to submit the work for publication.

### Author contributions

Sabhya Rana, Conceptualization, Data curation, Formal analysis, Validation, Investigation, Visualization, Methodology, Writing – original draft, Project administration, Writing – review and editing; Firoj Alom, Data curation, Formal analysis, Validation, Investigation, Visualization, Methodology, Writing – original draft, Writing – review and editing; Robert C Martinez, Data curation, Investigation, Visualization; David D Fuller, Conceptualization, Supervision, Funding acquisition, Investigation, Methodology, Writing – original draft, Project administration, Writing – review and editing; Aaron D Mickle, Conceptualization, Resources, Formal analysis, Supervision, Funding acquisition, Investigation, Visualization, Writing – original draft, Project administration, Writing – review and editing

### Author ORCIDs

Sabhya Rana https://orcid.org/0000-0002-1303-6614
Firoj Alom http://orcid.org/0000-0002-7498-006X
Aaron D Mickle https://orcid.org/0000-0002-6454-8038

### Ethics

All procedures were approved by the Institutional Animal Care and Use Committee at the University of Florida and are in accordance with the National Institutes of Health Guidelines (Protocol # 202107438). Animals were housed individually in cages under a 12-hr light/dark cycle with ad libitum access to food and water.

Reviewer #1 (Public Review): https://doi.org/10.7554/eLife.89767.3.sa1
Reviewer #2 (Public Review): https://doi.org/10.7554/eLife.89767.3.sa2
Reviewer #3 (Public Review): https://doi.org/10.7554/eLife.89767.3.sa3
Author Response https://doi.org/10.7554/eLife.89767.3.sa4

## Additional files

### Supplementary files
• Supplementary file 1. Mean data of cystometry measures at 5 d post-injury following HPCD or ampakine CX1739 treatment.

• Supplementary file 2. Mean data of EUS EMG activity at 5 d post-injury following HPCD or ampakine CX1739 treatment.

• MDAR checklist

### Data availability
All data for this study is publicly available at: https://doi.org/10.6084/m9.figshare.25193603.v1. All summary data has been provided in *Supplementary files 1 and 2*.

The following dataset was generated:

| Author(s) | Year | Dataset title | Dataset URL | Database and Identifier |
|---|---|---|---|---|
| Rana S, Alom F, Martinez RC, Fuller DD, Mickle AD | 2024 | Rana et al. 2024: Cystometry and EUS EMG Data Set | https://doi.org/10.6084/m9.figshare.25193603.v1 | figshare, 10.6084/m9.figshare.25193603.v1 |

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
