## [Editor Report · eLife assessment]

Bladder dysfunction following spinal cord injury (SCI) represents a severe and disabling complication without effective therapies. Following evidence that AMPA receptors play a key role in bladder function, the authors show **convincingly** that AMPA allosteric activators can ameliorate many of the subacute defects in bladder and sphincter function following SCI, including prolonged voiding intervals and high bladder pressure thresholds for voiding. These **valuable** results in rodents may help in the development of these agents as therapeutics for humans with SCI-induced bladder dysfunction.

---

## [Referee Report · Reviewer #1 (Public Review)]

Summary:

Spinal cord injury (SCI) causes immediate and prolonged bladder dysfunction, for which there are poor treatments. Following up on evidence that AMPA glutamatergic receptors play a key role in bladder function, the authors induced spinal cord injury and its attendant bladder dysfunction and examined the effects of graded doses of allosteric AMPA receptor activators (ampakines). They show that ampakines ameliorate several prominent derangements in bladder function resulting from SCI, improving voiding intervals and pressure thresholds for voiding and sphincter function.

Strengths:

Well performed studies on a relevant model system. The authors induced SCI reproducibly and showed that they had achieved their model. The drugs revealed clear and striking effects. Notably, in some mice which had such bad SCI that they could not void, the drug appeared to restore voiding function.

Weaknesses:

The studies are well conducted, but it would be helpful to include information on the kinetics of the drugs used, their half-life and how long they are present in rats after administration. What blood levels of the drugs are achieved after infusion? How do these compare with blood levels achieved when these drugs are used in humans?

---

## [Referee Report · Reviewer #2 (Public Review)]

Summary:

In this study, Rana and colleagues present interesting findings demonstrating potential beneficial effects of AMPA receptor modulator with ampakines in the context of neurogenic bladder following acute spinal cord injury. Neurogenic bladder dysfunction is characterized by urinary retention and/or incontinence, with limited treatments available. Based on recent observations showing that ampakines improved respiratory function in rats with SCI, the authors explored the use of ampakine CX1739 on bladder and external urethral sphincter (EUS) function and coordination early after mid-thoracic contusion injury. Using continuous flow cystometry and EUS myography the authors showed that ampakine treatment led to decreased peak pressures, threshold pressure, intercontraction interval and voided volume in SCI rats versus vehicle-treated controls. Although CX1739 did not alter EUS EMG burst duration, treatment did lead to EUS EMG bursting at lower bladder pressure compared to baseline. In a subset of rats that did not show regular cystometric voiding, CX1739 treatment diminished non-voiding contractions and improved coordinated EUS EMG bursting. Based on these findings the authors conclude that ampakines may have utility in recovery of bladder function following SCI.

Strengths and Weaknesses:

The experimental design is thoughtful and rigorous, providing evaluation of both the bladder and external urethral sphincter function in the absence and presence of ampakine treatment. The data in support of a role for CX1789 treatment in the context of neurogenic bladder are presented clearly, and the conclusions are adequately supported by the findings. The authors have addressed essentially all of the weaknesses related to translational significance, CX1789 half-life, and the use of female animals only in this study.

---

## [Referee Report · Reviewer #3 (Public Review)]

Summary:

In this manuscript, Rana and colleagues examined the effect of a "low impact" ampakine, an AMPA receptor allosteric modulator, on the voiding function of rats subjected to midline T9 spinal cord contusion injury. Previous studies have shown that the micturition reflex fully depends on AMPA glutaminergic signaling, and, that the glutaminergic circuits are reorganized after spinal cord injury. In chronic paraplegic rats, other circuits (no glutaminergic) become engage in the spinal reflex mechanism controlling micturition. The authors employed continuous flow cystometry and external urethral sphincter electromyography to assess bladder function and bladder-urethral sphincter coordination in naïve rats (control) and rats subjected to spinal cord injury (SCI). In the acute phase after SCI, rats exhibit larger voids with lower frequency than naïve rats. This study shows that CX1739 improves, in a dose-dependent manner, bladder function in rats with SCI. The interval between voids and the voided volume were reduced in rat with SCI when compared to controls. In summary, this is an interesting study that describes a potential treatment for patients with SCI.

Strengths:

The findings described in this manuscript are significant because neurogenic bladder predisposes patients with SCI to urinary tract infections, hydronephrosis and kidney failure. The manuscript is clearly written. The study is technically outstanding, and the conclusions are well justified by the data.

Weaknesses:

The study was conducted 5 days after spinal cord contusion when the bladder is underactive. In rats with chronic SCI, the bladder is overactive. Therefore, the therapeutic approach described here is expected to be effective only in the underactive bladder phase of SCI. The mechanism and site of action of CX1739 is not defined.

---

## [Author Response]

The following is the authors’ response to the original reviews.

Response to Reviewing Editor:Comment: Bladder dysfunction following spinal cord injury (SCI) represents a severe and disabling complication and we lack effective therapies. Following evidence that AMPA receptors play a key role in bladder function the authors show convincingly that AMPA allosteric activators can ameliorate many of the subacute defects in bladder and sphincter function following SCI, including prolonged voiding intervals and high bladder pressure thresholds for voiding. These valuable results in rodents may help in the development of these agents as therapeutics for humans with SCI-induced bladder dysfunction.

Response: We thank the reviewing editor for their assessment of this manuscript and positive comments. We also appreciate the opportunity to revise this manuscript for publication in eLife. We have addressed the excellent comments of the three reviewers. We have included detailed response-to-reviewer comments below to address each specific point. Based on the reviewers’ critiques, we feel our re-working of the manuscript has made for a greatly improved study.

**Public Reviews:**

**Reviewer #1 (Public Review):**
Summary:Spinal cord injury (SCI) causes immediate and prolonged bladder dysfunction, for which there are poor treatments. Following up on evidence that AMPA glutamatergic receptors play a key role in bladder function, the authors induced spinal cord injury and its attendant bladder dysfunction and examined the effects of graded doses of allosteric AMPA receptor activators (ampakines). They show that ampakines ameliorate several prominent derangements in bladder function resulting from SCI, improving voiding intervals and pressure thresholds for voiding and sphincter function.Strengths:Well-performed studies on a relevant model system. The authors induced SCI reproducibly and showed that they had achieved their model. The drugs revealed clear and striking effects. Notably, in some mice that had such bad SCI that they could not void, the drug appeared to restore voiding function.Weaknesses:The studies are well conducted, but it would be helpful to include information on the kinetics of the drugs used, their half-life, and how long they are present in rats after administration. What blood levels of the drugs are achieved after infusion? How do these compare with blood levels achieved when these drugs are used in humans?

Response: We thank Reviewer #1 for the positive comments and their helpful critique. We address each of the specific comments below (in the “Recommendations for the Authors” section of this Response to Reviewer Comments document), and have made changes to the manuscript based on these excellent points.

**Reviewer #2 (Public Review):**
Summary:In this study, Rana and colleagues present interesting findings demonstrating the potential beneficial effects of AMPA receptor modulators with ampakines in the context of the neurogenic bladder following acute spinal cord injury. Neurogenic bladder dysfunction is characterized by urinary retention and/or incontinence, with limited treatments available. Based on recent observations showing that ampakines improved respiratory function in rats with SCI, the authors explored the use of ampakine CX1739 on bladder and external urethral sphincter (EUS) function and coordination early after mid-thoracic contusion injury. Using continuous flow cystometry and EUS myography the authors showed that ampakine treatment led to decreased peak pressures, threshold pressure, intercontraction interval, and voided volume in SCI rats versus vehicle-treated controls. Although CX1739 did not alter EUS EMG burst duration, treatment did lead to EUS EMG bursting at lower bladder pressure compared to baseline. In a subset of rats that did not show regular cystometric voiding, CX1739 treatment diminished non-voiding contractions and improved coordinated EUS EMG bursting. Based on these findings the authors conclude that ampakines may have utility in recovery of bladder function following SCI.Strengths:The experimental design is thoughtful and rigorous, providing an evaluation of both the bladder and external urethral sphincter function in the absence and presence of ampakine treatment. The data in support of a role for CX1789 treatment in the context of the neurogenic bladder are presented clearly, and the conclusions are adequately supported by the findings.Weaknesses:Since CX1789 was administered in the context of cystometry and urethral sphincter EMG, a brief discussion of how ampakines could be used in a therapeutic context in humans would help to understand the translational significance of the work. The study lacks information on the half-life of CX1789 and how might this impact the implementation of CX1789 for clinical use. In addition, the study was limited to female rats. Lastly, given the male bias of traumatic SCI in humans, a brief discussion of this limitation is warranted.

Response: We thank Reviewer #2 for their positive comments and their helpful critique. We address each of the specific comments below (in the “Recommendations for the Authors” section of this Response to Reviewer Comments document). We have also made changes to the manuscript based on the three excellent discussion points brought up by the reviewer.

**Reviewer #3 (Public Review):**
Summary:In this manuscript, Rana and colleagues examined the effect of a "low impact" ampakine, an AMPA receptor allosteric modulator, on the voiding function of rats subjected to midline T9 spinal cord contusion injury. Previous studies have shown that the micturition reflex fully depends on AMPA glutaminergic signaling, and, that the glutaminergic circuits are reorganized after spinal cord injury. In chronic paraplegic rats, other circuits (no glutaminergic) become engaged in the spinal reflex mechanism controlling micturition. The authors employed continuous flow cystometry and external urethral sphincter electromyography to assess bladder function and bladder-urethral sphincter coordination in naïve rats (control) and rats subjected to spinal cord injury (SCI). In the acute phase after SCI, rats exhibit larger voids with lower frequency than naïve rats. This study shows that CX1739 improves, in a dose-dependent manner, bladder function in rats with SCI. The interval between voids and the voided volume was reduced in rats with SCI when compared to controls. In summary, this is an interesting study that describes a potential treatment for patients with SCI.Strengths:The findings described in this manuscript are significant because neurogenic bladder predisposes patients with SCI to urinary tract infections, hydronephrosis, and kidney failure. The manuscript is clearly written. The study is technically outstanding, and the conclusions are well justified by the data.Weaknesses:The study was conducted 5 days after spinal cord contusion when the bladder is underactive. In rats with chronic SCI, the bladder is overactive. Therefore, the therapeutic approach described here is expected to be effective only in the underactive bladder phase of SCI. The mechanism and site of action of CX1739 is not defined.

Response: We thank Reviewer #3 for the positive comments and their helpful critique. We address each of the specific comments below (in the “Recommendations for the Authors” section of this Response to Reviewer Comments document), and have made changes to the manuscript based on the excellent point mentioned in the weakness section.

Comment: Recommendations for the authors: please note that you control which revisions to undertake from the public reviews and recommendations for the authors

Response: We have addressed all comments of both reviewers. We detail our responses in this Response to Reviewer Comments document and have made the associated modifications to the revised manuscript.

**Reviewer #1 (Recommendations For The Authors):**
Comment: These are well-performed studies.

Response: We thank the reviewer for their positive comment.

Comment: It would be useful to know the blood levels of the drug that are achieved by the infusions, and how long the drugs remain after infusion. Is the 45-minute interval between doses appropriate for the drug's kinetics?

Response: While blood levels of ampakine were not tested in this study, pharmacokinetic parameters for CX1739 in Sprague Dawley rats have previously been determined following an intravenous administration of CX1739. The mean plasma half-life of CX1739 was 1.25 ± 0.03 hrs, with a Tmax of 30 minutes (information provided through personal communication with RespireRx). Although the 45 minutes interval between doses would not be within the time frame of post administration clearance of the first CX1739 dose from the system, the plasma levels would be considerably lower by 45 mins post administration. A limitation of terminal cystometry preparations is the duration you can maintain a single animal, and this was also included in our rationale for dosing every 45 mins. In our experience longer recordings can increase variability. A 45 min window allowed for the anesthetized procedure to remain under ~6 hours. Further, in our studies investigating the impact of ampakines in rats following an SCI, acute impacts of intravenous ampakine administration were observed for up to 30 minutes. (Rana et al., 2021) Along with the half-life and data from the respiratory system informed our decision here. We have added this rationale to the methods section and in part to the discussion section (Page 11, 2930).

Comment: Since a major plus of these studies is their potential applicability to humans with SCI, it would be helpful to know whether the drug levels achieved here resemble those that were achieved in human trials to date.

Response: Since blood/plasma levels were not tested in the current study, we cannot comment on the comparison of blood plasma levels achieved in human trials. However, we have expanded upon this point in the discussion section (page 29-30).

Comment: The authors could also provide us with a bit more description of the different classes of ampakines, and why they chose the one they used.

Response: Thank you for this suggestion. We would like to highlight a section in our discussion (Page 28-29) where we have an in-depth description of the two classes of ampakines in the discussion and the rationale for selecting the low-impact CX1739 drug.

Comment: Lastly, the first reference is cited twice in the bibliography.

Response: The duplicate reference has been removed.

**Reviewer #2 (Recommendations For The Authors):**
Comment: Overall, the findings support the potential for ampakine administration in the setting of neurogenic bladder dysfunction following SCI. The manuscript was well written, the experimental design was rigorous, the data were of excellent quality, and the conclusions were adequately supported by the findings. Weaknesses are considered minor and can be addressed mostly by clarification as noted below.

Response: We thank the reviewer for their positive comments.

Comment: Since CX1789 was provided in the context of cystometry and EUS EMG, a brief discussion of how ampakines could be used in a therapeutic context in humans would help to understand the translational significance of the work.

Response: Thank you for this important comment to include a discussion about translational significance of CX1739. We have included a discussion (Page 34) about the translational significance of this work in the discussion section of the last paragraph.

Comment: No information is provided on the half-life of CX1789 and how might this impact the implementation of CX1789 for clinical use. The inclusion of this information would help the reader to appreciate the potential for and limitations of clinical implementation.

Response: Although pharmacokinetic analyses were not conducted as part of this study, we have included details of CX1739 plasma pharmacokinetics examined in Sprague-Dawley rats (Page 11, 29-30). This information has been provided through personal communication with RespireRx.

Comment: The study was limited to female rats. Would the authors anticipate different efficacy of CX1789 in male rats? A comment on the choice of animal sex and implications for interpretation of the findings would strengthen the discussion and potential clinical implementation given the male bias of traumatic SCI in humans.

Response: Thank you for your important comment. In this study, females were chosen primarily due to the fact they have better recovery outcomes from spinal cord injury. During initial preliminary data gathering, we used both male and female rats and found that the male rats often did not recover cytometric voiding at this time point. So we chose to continue only with the female rats in this current study. It is well established that female rats have better urogenic recovery from SCI effects, perhaps due to the easier postoperative care. It is critical that we complete future studies in both male and female rats, however, we will have to change our experimental paradigm (time after injury, and or severity of injury) to make comparisons between SCI and intact male rats. We have now included this important topic of our sex selection in the methods section (Page 6) of the manuscript and have also expanded this point in the discussion section (page 30).

**Reviewer #3 (Recommendations For The Authors):**
Comment: The impact of ampakine treatment on EUS EMG activity is not obvious from the data presented in Fig. 5C-F. I do see in the magnified area of the SCI rat tracing some clear EUS activity with 15 mg/kg of CX1739. However, statistically, there is not a significant improvement in bladder-urethral sphincter coordination in rats treated with ampakine. Authors should discuss how or why ampakine treatment improves bladder function without affecting bladder-urethral sphincter coordination. The background noise of the EUS EMG in Fig. 5B changes dramatically between conditions. Are these tracings from the same experiment? If yes, please explain why the background noise changes during the course of the experiment. Was this change in background noise observed only in SCI rats?

Response: Thank you for such an interesting comment. Although our data analysis shows no statistically significant difference in the duration or amplitude of EUS EMG bursting when comparing vehicle to ampakine treatment. However, we did see a difference in the threshold at which bursting occurred (Fig 5C-F). Rats that lost complete coordination (Figure 6) due to injury, ampakines provide further confirmation about producing EUS EMS bursting and coordinated voiding.

Therefore, these results suggest that ampakines have some positive modulatory effects on EUS EMG bursting events. Overall, we did not see any significant differences of the background noise of EUS EMG between conditions during experiments both in spinal intact and SCI. The background noise of the EUS EMG in Fig. 5B decreases after baseline and HPCD due to changes in experimental conditions (needed to use slightly more urethane due to showing up of animal’s consciousness). We would also like to confirm that these tracings are from the same experiment. Accordingly, we have made further clarifications in the manuscript.

Comment: Tables 1 and 2 show the same data as figures 3 and 4. I suggest removing the tables. In addition, table 2 includes letters (A, B, C, D) to indicate statistical significance. However, no indication of the meaning of these letters is provided. What does "levels not connected by same letter are significantly different" mean? Please clarify. I suggest including the statistical comparisons in Fig. 4

Response: While we did consider adding statistical bars in the graphs themselves, the number of comparisons being conducted reduced the readability of the graphs. Thus, we would like preserve the current format of the table and provide the readers with all statistical comparisons being made. The statement “levels not connected by the same letter are significantly different” indicates that only treatment groups for an outcome that do not have an overlapping letter, such as baseline (A) and HPCD (A) values for threshold pressures are different from the 5 mg/kg (B,C,D), 10 mg/kg (C,D) and 15 mg/kg (D) group in the SCI rats. Further, threshold pressures in the 5 mg/kg, 10 mg/Kg and 15 mg/kg groups are not significantly different from each other. These results have also been described in detail in the results section. Lastly, we acknowledge the redundancy of data presented in Tables 1 and 2. These two tables have been moved to the supplemental section.

Comment: A study by Yoshiyama and colleagues previously showed that the AMPA antagonists LY215490 completely abolished the reflex bladder contractions and EMG activity of the EUS muscle during a continuous filling in naïve rats (JPET 1997). Surprisingly, CX1739, a low-impact AMPA receptor activator, does not affect bladder contractions or EMG activity in naïve rats. Authors should discuss the reason for this discrepancy.

Response: Thank you for this comment. We believe the different pharmacokinetics of the drugs can explain these effects. We have included this critical point in the discussion (page 31-32).

Comment: The conclusion that CX1739 is acting on sensory pathways is highly speculative and needs additional support. The functional status of the afferent pathways is uncertain following SCI. Please revise.

Response: Thank you for this comment. We agree, in retrospect, that this speculative comment is an overassumption, and we have removed it from the discussion. We have modified the discussion to remove focus from the sensory nervous system and, more generally, discuss the location of AMPA receptors in the voiding neurocircuitry (page 31).

Comment: Figure 3. It's difficult to see the asterisks that indicate statistical significance. Please use a line or a bigger symbol to indicate statistical differences between groups.

Response: Thank you for the suggestion we have modified the figure to make the asterisks bigger and added a line.

Comment: Data for peak pressure should be included in Figures 3 and 4.

Response: Thank you for pointing out one of the important parameters of cystometry which is peak pressure. As we did not see significant changes in bladder peak contraction pressure between spinal intact and SCI rats, we prefer not to show a graph of peak pressure (in Fig 3) to highlight other parameters that showed significant injury effects, such as baseline pressure, ICI, threshold, and voided volume. However, peak pressure reduced similarly both in spinal intact and SCI rats, suggesting that ampakine has some treatment effects on peak pressure that we prefer to include in Fig 4. We modified our results section and have included a description on peak pressures in the result section.

Comment: The peak pressure was reduced in both naïve and SCI rats treated with ampakine. Therefore, the peak pressure is not one of the parameters that improves by ampakine in SCI rats.

Response: Yes, we agree that peak pressures between spinal intact and SCI rats were comparable. Some treatment effects of ampakine on peak pressure were observed both between spinal intact and SCI rats. We have amended the manuscript to make this clearer.

Comment: The reference from Yoshiyama et al (1999) is duplicated.

Response: Thank you for catching this error. The references have been combined in the revised version.

Comment: Page 15, the authors state that "Coordinated bladder contractions and associated EUS EMG activity were readily demonstrated in all 7 naïve animals". In other sections, they referred to 8 naïve rats. What is the actual number of naïve rats?

Response: Thanks for pointing out this error. The actual number of naïve rats is 8. We have rectified this error.